# Sonochemical Synthesis, Characterization and Optical Properties of Tb-Doped CdSe Nanoparticles: Synergistic Effect between Photocatalysis and Sonocatalysis

**DOI:** 10.3390/nano11020378

**Published:** 2021-02-02

**Authors:** Younes Hanifehpour, Narges Nozad Ashan, Ali Reza Amani-Ghadim, Sang Woo Joo

**Affiliations:** 1Department of Chemistry, Sayyed Jamaleddin Asadabadi University, Asadabad 6541861841, Iran; 2Office of Management Development and Research, East Azarbaijan’s Water and Wastewater Company, Tabriz P.O. Box 83714-161, Iran; nozad2484@gmail.com; 3Applied Chemistry Research Laboratory, Department of Chemistry, Faculty of basic Science, Azarbaijan Shahid Madani University, Tabriz P.O. Box 83714-161, Iran; amani.gh@azaruniv.ac.ir; 4New Technologies in the Environment Research Center, Azarbaijan Shahid Madani University, Tabriz P.O. Box 83714-161, Iran; 5School of Mechanical Engineering, Yeungnam University, Gyeongsan 712-749, Korea

**Keywords:** sono-photocatalytic degradation, terbium, response surface methodology, synergistic index

## Abstract

In this study, Tb-doped CdSe nanoparticles with variable Tb^3+^ content were synthesized by a simple sonochemical technique. The synthesized nanoparticles were characterized by X-ray photoelectron spectroscopy (XPS), scanning electron microscopy (SEM), and powder X-ray diffraction (XRD). The sono-photocatalytic activities of the as-prepared specimens were assessed for the degradation of Reactive Black 5. The experimental results show that the sono-photocatalytic process (85.25%) produced a higher degradation percentage than the individual sono- (22%) and photocatalytic degradation (8%) processes for an initial dye concentration and Tb-doped CdSe dosage of 20 mg/L and 1 g/L, respectively. Response surface methodology (RSM) was utilized to assess model and optimize the impacts of the operational parameters, namely, the Tb^3+^ content, initial dye concentration, catalyst dosage, and time. The addition of benzoquinone results in remarkably inhibited degradation and the addition of ammonium oxalate reduced the removal percentage to 24%. Superoxide radicals and photogenerated holes were detected as the main oxidative species.

## 1. Introduction

Degrading hazardous organic contaminants in industrial wastewater through AOPs (advanced oxidation processes) has attracted a huge deal of research attention. Production of •OH radicals is the primary mechanism of AOPs, which have higher oxidization potential and can help to obtain efficient and faster degradation of the pollutants [1,2]. The AOP process is suitable particularly for cleaning up non-degradable or toxic materials, like petroleum constituents, pesticides, volatile organic compounds in wastewater, and aromatics [3,4].

Sonocatalysis is one type of AOP process that has recently been applied for degrading organic dyes [4]. The chemical influence of ultrasound (US) is caused by the acoustic cavitation leading to the implosive collapse growth, and formation of bubbles within a solution. By collapsing the bubbles, localized hot spots are produced with very high pressure and temperature. By such severe circumstances, the water molecules and dissolved oxygen can experience direct thermal dissociation to create highly reactive radical species like •OH, oxygen (•O), and hydrogen (•H) playing a key role to oxidize organic pollutants in water [5,6,7]

The combination of sonocatalysis and photocatalysis appears to improve the degradation share of organic pollutants owing to synergistic effects [8,9]. In addition to the formation of reactive oxygen species (ROS) via cavitation, the use of US in photocatalytic processes has other benefits, such as incrementing the active surface area of the catalyst by preventing the aggregation of particles, increasing the pollutants’ mass transfer between the solution and catalyst surfaces, preventing catalyst deactivation by continuously cleaning absorbed molecules from the catalyst surface by micro streaming and micro bubbling, and incrementing the number of high pressure and temperature regions by breaking up microbubbles made via the US into smaller ones in the presence of catalyst particles [10,11,12,13].

CdSe is an II−VI semiconducting material that has been investigated extensively as a result of its small band gap energy (1.65 to 1.8 eV) [14,15,16]. It is appropriate for different optoelectronic uses in catalysis, biological labeling, solar cells, etc. [17,18,19,20]. The photocatalytic application of CdSe has been restricted due to its narrow bandgap and subsequently high recombination probability [21,22]. However, the fast recombination of photogenerated electron-holes restricts its practical application because the photoinduced electrons and holes are neutralized before they can initiate the photocatalytic processes. Lanthanide-doped nanostructured semiconductors have recently been used as an active photocatalyst and sonocatalyst [23,24,25]. Rare earth cations with empty 5d orbitals and partly occupied 4f orbitals could also notably enhance the separation rate of photo-induced charge carriers within semiconductor sonocatalysts and improve significantly the sonocatalytic activity [26,27].

The sono and photocatalytic application of pure CdSe nanoparticles has been rarely reported due to its narrow bandgap. Therefore, this work deals with introducing a simple sonochemical path for the synthesizing pure and terbium-doped CdSe (Tb*_x_*Cd_1−*x*_Se) nanoparticles. The nanoparticles’ sonocatalytic activity was assessed for RB5 (as a model organic dye) (Table 1). No report exists on the sonocatalytic degradation of Reactive Black 5 (RB5) in the existence of Tb-doped CdSe nanoparticles. The sono-photocatalytic degradation of organic dye was modeled and enhanced by response surface methodology (RSM), and the effects of inorganic ions on the degrading efficacy of RB5 was investigated.

## 2. Experimental Methods

### 2.1. Chemicals and Materials

The chemicals utilized in this work had analytical grades and were utilized with no further purification. Se (99.99%), Tb (NO_3_)_3_·5H_2_O, ethanol (99%), and NaOH were attained from Sigma-Aldrich. Cd (NO_3_)_2_·4H_2_O (99.5%) and N_2_H_4_·H_2_O (99%) were purchased from Merck, and RB5 was obtained from the Zhejiang Yide Chemical Company (China).

### 2.2. Characterization

The samples’ crystal phase composition was determined using XRD characterization at room temperature through a D8 Advance diffractometer (Bruker, Karlsruhe, Germany) with monochromatic high-intensity Cu K*a* radiation (λ = 1.5406 Å), an emission current of 30 mA, and accelerating voltage of 40 kV. Using an electron microscope (SEM, S-4200, Hitachi, Tokyo, Japan), the morphologies of synthesized samples were observed. The chemical state and chemical composition were defined using X-ray photoelectron spectroscopy (XPS) (K-Alpha, Thermo Fisher, Waltham, MA, USA). A diffuse reflectance UV–VIS spectrophotometer was utilized to obtain the samples’ optical absorption spectra (Varian Cary 3 Bio, Varian Ltd., Artarmon, Australia).

### 2.3. Preparation of Tb-Doped CdSe Nanoparticles

Tb-doped cadmium selenide compounds with different Tb contents (0–12 mol%) were made sonochemically utilizing hydrazine hydrate (N_2_H_4_·H_2_O) as a reducing agent. Within a characteristic synthesis, Cd (NO_3_)_2_·4H_2_O, proper molar ratios of Tb (NO_3_)_3_·5H_2_O, 2 mmol of Se powder, and 1 mmol of NaOH were dissolved first in 50 mL of distilled water. Next, drops of hydrazine hydrate (N_2_H_4_·H_2_O) were inserted into the solution with stirring at moderate speed. Lastly, sonicating the mixture was performed for 2 h using a bath-type sonicator (SW12H, Fisher Scientific, Loughborough, UK) with a frequency of 37 kHz and output intensity of 200 W. Gathering the as-prepared Tb_x_Cd_1−x_Se nanoparticles, they were rinsed numerous times with absolute ethanol and distilled water for removing residual impurities and then dried at vacuum for 3 h at 50 °C. The results were a black powder.

### 2.4. Assessing Catalytic Activity

The sonophotocatalytic activities of the Tb-doped and undoped CdSe nanoparticles were assessed to decolorize RB5 as a model dye contaminant. Within a characteristic procedure, suspending the nanocatalyst (0.1 g) was performed in an aqueous solution of the model dye (100 mL) with an identified initial concentration. Then, the suspension was irradiated by a 40 W compact fluorescent visible light lamp armed with a cutoff filter to present visible light illumination (λ of higher than 420 nm) into the ultrasonic bath. A UV–VIS spectrophotometer was utilized to determine the removal of dye via the absorbance at λ_max_ = 597 nm. The decolorization efficiency (*DE*) was determined as follows:(1)$$DE\;\left(\%\right)=\left(1-\frac{C}{C_{0}}\right)\times 100$$
where *C* and *Co* are the dye’s final and initial concentrations within the solution (mg/L), respectively. To test the nanocatalyst’s reusability, the utilized catalyst was separated from the solution, rinsed with distilled water and utilized in a fresh test after drying at 50 °C.

### 2.5. Experimental Design

RSM is a collection of statistical and mathematical methods that are used to model and optimize a process. Compared to the typical “one factor at a time” method, RSM can decrease the number of experiments required to assess the impacts of operating parameters. Other advantages are the ability to examine the impacts of different variables and their interactions on the output response. Statistical techniques for experimental design can lead to reduced process variability reducing the time, reagents, and experimental work required [28,29].

Four factors were considered as the input variables. Table 2 shows the experimental ranges in actual and coded values. The coded values of variables were determined as follows:(2)$$Coded\;value=\frac{x_{i}-x_{0}}{\Delta x}$$
where *x_i_*, *x*_0_, and Δ*x* are the variable’s coded level (dimensionless value), the variable’s center point, and the interval variation, respectively.

The extensively used Central Composite Design (CCD) methodology was utilized to evaluate the interactive and individual impacts of 4 main variables on the decolorization efficiency response. In total, 31 experiments were carried out, as shown in Table 3. These consisted of 8 axial points (or star points coded as ±α, where α = 2), 2^4^ orthogonal two-level full factorial design points (coded as ±1), and 7 replications of the central points. The experiment design was performed using Design Expert 7 software (7.3.1, Stat-Ease, Minneapolis, MN, USA).

## 3. Results and Discussion

### 3.1. Characterizing the Synthesized Samples

The P-XRD (powder X-ray diffraction) patterns of the Tb-doped CdSe and pure samples are shown in Figure 1. The specimens diffraction peaks are indexed readily to pure characteristic well-crystallized hexagonal CdSe ((No. 08-0549, a = 0.4299 nm, space group P63mc, and c = 0.7010 nm) [30,31,32]. There was no peak revealing impurities, which confirmed the effectiveness of the sonochemical route to synthesize the preferred specimens. Furthermore, the sharp diffraction peaks in the XRD spectra show that the as-prepared compound was highly crystalline. There were further unknown phases at doping levels of *x* = 0.12 for Tb^3+^. A slight shift was found to higher diffraction angles in the 12% Tb-doped CdSe patterns. This could be associated with the CdSe lattice’s contraction owing to the incidence of Tb^3+^ions, with a smaller radius (0.92 Å) in comparison to Cd^2+^ions (0.97 Å).

SEM analyses were done to clarify the size and shape of the nanoparticles. Figure 2 and Figure 3 show the SEM microphotographs of the CdSe and Tb-doped CdSe samples, respectively. Compared to the pure CdSe, the SEM images show a larger particle size of the terbium-doped CdSe nanoparticles. This proves that the doping of Tb^3+^ions into the CdSe lattice increases the aggregation of nanoparticles and correspondingly increases the size of the particles.

Figure 4a,b shows that the size distribution of the as-synthesized Tb_1−*x*_Cd_*x*_Se compound is in the range of 30 to 40 nm which is larger than that of the undoped CdSe nanoparticles (20–30 nm). Such figures denote that the CdSe nanoparticles’ morphology is not changed by the doping of Tb^3+^ into the CdSe structure.

XPS analysis was carried out to prove the Tb ions incorporation into the CdSe crystal lattice and to evaluate the oxidation state of terbium. The narrow scan XPS and XPS spectrum of Tb_0.08_Cd_0.92_Se nanoparticles are shown in Figure 5a–d. The narrow scan spectra of Cd 3d in Figure 5b shows two peaks centered at 412 and 405 eV, which can be associated with the transition of respectively Cd 3d3/2 and Cd 3d5/2 [30]. The single peak placed at a binding energy of 54.58 eV is associated with the Se 3d transition (see Figure 5c) [31]. As shown in Figure 5d, the Tb 3d5/2 and 3d3/2 peaks placed at 1276.49 and 1243.56 eV confirm that Tb ions were doped successfully into the CdSe’s crystal lattice [33].

The UV–VIS diffuse reflectance spectrum was used to study the optical absorption property, as shown in Figure 6. To quantify the shift of the absorption edge, the band-gap energies of Tb-doped CdSe and pure CdSe were calculated by Tauc’s equation.

Figure 7 represents the Tauc plot of (*hvα*)^2^ vs. (*hv*). The as-prepared compound’s bandgap energy can be determined from the interception of the resultant linear area with the energy axis. The doped CdSe has lower Eg values compared to the pure sample, and it is reduced by increasing the dopant. The bandgap energy of pure and Tb-doped CdSe is given in Table 4.

### 3.2. Synergistic Effects of Sonocatalysis and Photocatalysis on the Degradation of RB5 Using Tb-doped CdSe

Through some comparative tests, the sono-, photo-, and sono-photocatalytic behavior of the synthesized Tb-doped CdSe were determined, for which the findings are presented in Figure 8a. The decolorization efficiency was negligible in the absence of catalyst particles under light irradiation, which shows that photolysis does not contribute to the RB5 removal from the aqueous solution. The results of the test conducted in dark conditions revealed that surface adsorption has no significant effect on decolorizing the dye solution. The removal percentage by the photocatalytic procedure was less than 10%. The decolorization efficiency by sonocatalytic degradation (22%) was greater than that of the sonolysis process (10%). The decolorization efficiency was significantly improved in the sono-photocatalytic process (87%) as a result of synergistic impacts between sono- and photocatalysis. This can be summarized as follows: first, the generation of several ROSs through the integrated photocatalytic procedure and cavitation effects; second, the improved mass transfer rates; third, the catalyst particles disaggregation through the US and the larger active surface areas; and fourth, the creation of more hot spots by existing catalyst particles [34].

As demonstrated in Figure 8b, the plot of (−ln (*C*/*C*_0_) vs. time shows linear dependence in the case of all three photocatalytic, sonocatalytic, and sono-photocatalytic processes, indicating pseudo-first-order kinetics. The plot of (−ln (*C*/*C*_0_) vs. time was drawn for an initial RB5 concentration range of 10–30 mg/L, and the results are given in Table 5. It is obvious that the decolorization efficiency and pseudo-first-order kinetic constant in sono-photocatalysis (k_obs,sono-photo_) are greater than those for photocatalysis (k_obs,photo_) and sonocatalysis (k_obs,sono_). These results implicitly include the concept of a synergistic effect. We define the synergistic index (SI) by the following equation:(3)$$\mathrm{SI}=\frac{k_{\mathrm{obs},\;\mathrm{sono}-\mathrm{photo}}}{k_{\mathrm{obs},\;\mathrm{photo}}+k_{\mathrm{obs},\mathrm{sono}}}$$

The obtained SI and pseudo-first-order kinetic constants for sonocatalysis, photocatalysis, and sono-photocatalysis are given in Table 5.

To assess the degradation process’s mechanism and to discover the main oxidative species, experiments were conducted in the existence of appropriate scavengers of active species. According to Figure 9, adding t-BuOH (a scavenger of hydroxyl radicals) leads to a reduction of 18% in the decolorization percentage. By adding oxalate (a scavenger of h^+^_VB_), the decolorization percentage decreased to 24%. When benzoquinone (BQ) was added (a scavenger of superoxide radicals), the dye degradation was inhibited remarkably. Such findings indicate that the h^+^_VB_ and superoxide radicals were the main oxidative species in degrading dye molecules. Though, the hydroxyl radicals also affect the decolorization. Regarding the synergistic impacts of sono- and photocatalysis, a possible mechanism for the degradation procedure can be suggested as follows:

(1) Both US and light irradiation can excite the catalyst nanoparticles to form electron-hole pairs:Tb-doped CdSe + (*hν* and *US*) *→* h^+^ + e^−^(4)

The formation of electron-hole pairs in semiconducting materials is very clear. However, the US radiation cannot excite the electrons from the valence band to the conduction band of a semiconductor. It is indicated that under the US radiation in the presence of solid oxides three processes can promote the efficiency. Firstly, the presence of semiconducting oxides promotes the formation of nuclei and subsequently increases cavitation and production of hydroxyl radicals, i.e., heterogenous nucleation. Moreover, free radicals generated via collapsing of microbubbles are recombined and emit high wavelengths of light with low energy. The mentioned phenomenon is known as sonoluminescence (SL). These emmitted wavelengths can excite the valance band electrons toward the conduction band where the bandgap of the semiconductor is consisting of the photons emitted through sonoluminescence (PSL). As the Tauc plot revealed, the bandgap of CdSe is considerably decreased by Tb doping which makes it susceptible to absorb the PSL. Accordingly, more active radicals will be produced through the reduction of dissolved oxygens by electrons in CB and/or oxidation of water by holes in VB. The third mechanism involves exploiting the local hot spots produced during ultrasonication. These hot spots can excite the semiconductor thermally leading to the production of electron-hole pairs [34,35]. The proposed mechanisms are given in Scheme 1.

(2) The conduction-band electrons can have reaction with adsorbed oxygen molecules to create ^•^O_2_^−^, HO_2_^•^, and H_2_O_2_ [36]:e^−^ + O_2_ → ^•^O_2_^−^(5)
H^+^ + ^•^O_2_^−^ → HO_2_^•^(6)
HO_2_^•^ + e^−^ + H^+^ → H_2_O_2_(7)

(3) Hydroxyl anions or water molecules can be oxidized by the photogenerated holes to form hydroxyl radicals [36]:h^+^ + OH^−^ → ^•^OH(8)
h^+^ + H_2_O → ^•^OH + H^+^(9)

(4) The water molecule pyrolysis can be promoted by the existence of ultrasonic irradiation to produce hydrogen and hydroxyl radicals [11]:H_2_O + US **→** H^•^ + ^•^OH(10)
^•^OH + ^•^OH → H_2_O_2_(11)

Ultimately, the dye molecules can be degraded by the created active spices:RB5 + ROSs → degrading dye molecules(12)

On the other hand, the presenese of Tb^3+^ cations in CdSe structure can relatively suppress the electron hole recombination. Tb^3+^ can take away the CB electrons and reduce to Tb^2+^ which will be oxidized by dissolved oxygen to Tb^3+^ due to the more stable electronic configuration of Tb^3+^ than Tb^3+^ producing very oxidative superoxide anion radical. Accordingly, the recombination of the photoexcited electrons and photogenerated holes was suppresed by the mentioned Red/Ox cycles (Scheme 1) [25,37].

### 3.3. Modeling of Sono-Photocatalytic Process by RSM and Investigation of the Impacts of Operational Factors

The second-order polynomial model that correlates the dependence of the removal efficiency on different parameters is given by the following equation [38]:(13)$$Y=b_{0}+\overset{n}{\underset{i=1}{\sum }}b_{i}x_{i}+\overset{n}{\underset{i=1}{\sum }}b_{ii}x_{i}^{2}+\overset{n}{\underset{i=1}{\sum }}\overset{n-1}{\underset{j=1}{\sum }}b_{ij}x_{i}x_{j}$$
where *y* denotes the predicted removal efficiency; *b*_0_ denotes the independent words; and *b_ij_*, *b_i_*, and *b_ii_* represent interaction, linear, and quadratic effects, respectively. The experimental results of degradation efficiency (Table 3) were fitted to the following equation with the coded variables:(14)$$y=66.62+2.48A+5.44B+14.78C-7.95D-6.05A^{2}-5.37B^{2}-4.88C^{2}-0.77D^{2}+0.56AB+1.02AC-2.71AD+0.46BC-1.29BD+0.043CD$$

The model’s significance and adequacy were assessed by analyzing variance (ANOVA), for which the findings are provided in Table 6. The importance of the presented model was revealed by a high Fisher’s F-value of 98.84 [29]. The probability values (*p*-value) less than 0.05 show that the null hypothesis is rejected, and the relevant co-efficient affects significantly the response. In contrast, the *p*-values higher than 0.05 indicate that the coefficient does not have an important influence on the output response [38]. The lack-of-fit *p*-value of the model was 0.31, denoting insignificant lack-of-fit for the obtained model. The high determination and adjusted regression coefficients (R^2^ = 98.8% and adj-R^2^ = 97.81%) show agreement between the observed and estimated responses and that the model can be effectively utilized to predict the removal efficiency.

The ANOVA results (Table 6), the values of the Student’s t distribution, and the related *p*-values (given in Table 7) show that the linear terms of all factors were significant. All of the quadratic terms were significant as well, except the initial dye concentration, which had a negligible effect on the output response. In the case of interactions of factors, only the interaction between Tb^3+^ and the initial dye concentration had a considerable effect, whereas interactive effects of other factors were insignificant. The insignificant terms with *p*-values higher than 0.05 can be removed from the model equation, and thus, the reduced form of the model is expressed by the following equation:$$y=66.62+2.48A+5.44B+14.78C-7.95D-6.05A^{2}-5.37B^{2}-4.88C^{2}-2.71AD$$

A Pareto analysis was performed to determine the percentage effect of each factor on the response [39]. Based on the results presented in Figure 10, the most important factor in the degradation efficiency was time (52.28%), followed by the initial dye concentration (15.3%), quadratic interaction of Tb^3+^ content (8.76%), and catalyst dosage (7.08%). In summary, the influence of terms on the degradation efficiency can be arranged in the following order:

Time > initial dye concentration > quadratic interaction of Tb^3+^ content > catalyst dosage ~ quadratic interaction of catalyst dosage > quadratic interaction of time.

Figure 11 shows the 3-D response surface and 2-D counter plots, which are the graphical representation of the estimated model equation and illustrate the effect of independent variables on the decolorization efficiency. Figure 11a displays the combined effect of the Tb^3+^ content and time, which revealed that the decolorization efficiency increased as the Tb^3+^ content increased from 0.02 to 0.06. As discussed in Section 3.1, the band gap energy of CdSe is decreased by doping with terbium. Incorporating Tb^3+^ ions into the CdSe lattice can introduce a narrow band inside the band gap of CdSe, resulting in a decrease in the band gap energy. As a consequence, the excitation of electrons from the valence band to the conduction band can happen more easily, and the decolorization efficiency increases. In addition, the Tb dopant can act as a scavenger for electrons and hence decrease the recombination rate of generated electron–hole pairs [40].

Although increasing the Tb^3+^ content further leads to a narrower band gap energy, it resulted in a decrease in the decolorization of the RB5 solution. The main reason for this decrease is that the recombination of photogenerated electron-hole pairs becomes easier because of three possible mechanisms. First, at high dopant concentration, the space charge layer can increase due to the light penetration resulting from US irradiation. Second, the large size of Tb ions compared to Cd ions limits their substitution into the lattice, and at high concentration, they can act as defect sites that stimulate recombination. Third, decreasing the distance between trapping sites decreases in the crystal structure at a high concentration of Tb^3+^ ions [40].

Figure 11b demonstrates the effect of the initial dye concentration. As the concentration of RB5 increased, the degradation percentage decreased gradually, which is typical in dye removal processes. For a given catalyst dosage, the number of active sites for the degradation of pollutant is constant, so the generation of oxidative species on the surface of the catalyst is reduced because of the coverage of these sites by dye molecules. Consequently, the removal efficiency decreases with an increase in the initial dye concentration.

Figure 11c shows the relationship between the catalyst dosage and decolorization efficiency. Increasing the catalyst dosage from 0.5 to 1 g/L increased the dye removal efficiency. At higher catalyst dosage, the surface area increases, and thus, there are more available active sites for the formation of hydroxyl radicals [36,40]. However, increasing the catalyst dosage further results in a decrease in the degradation percentage, which can be explained by the scattering of light and US waves because of the great number of catalyst particles, which limited the light penetration and heat and energy transmission near the surface of the catalyst [36].

### 3.4. Photostability and Photocatalyst Recycling

Figure 12 represents the reusability of 8% Tb-doped CdSe, which was tested by existing 1.12 g/L of catalyst powder, a dye concentration of 15 mg/L, and reaction time of 60 min. These conditions are the optimal values of operational parameters suggested by RSM. The estimated decolorization efficiency by RSM in the mentioned conditions was 87.53%, representing a decent consistency with the experimental efficiency (85.96%). After four repeated runs, the decrease in the sono-photocatalytic process was negligible, which confirms the synthesized nanoparticles’ reusability and high potential stability.

To compare our work with the previous studies, Appendix A compares the photocatalytic/sonocatalytic performance of prepared Tb-doped CdSe nanoparticles with that of the pristine, doped or some composites of CdSe in degradation of organic pollutants. It is clear that pristine CdSe exhibits no considerable efficienicy in degradation of organic pollutants due to the challenges discussed in Introduction section. Therefore, many attempts such as doping with different cations and preparation of nanocomposites were accomplished to enhance the CdSe activity. With respect to the mentioned table, Tb-doped CdSe provide considerable degradation efficiency compared with the other catalysts containing CdSe.

## 4. Conclusions

Through a simple sonochemical technique, Tb-doped CdSe nanoparticles were synthesized and successfully used as a visible-light-responsive sonophotocatalyst. Using the as-synthesized photocatalyst, the sono-photocatalytic decolorization was performed for RB5 in aqueous solution. The decolorization efficiency of the synthesized nanoparticles was much higher in the sono-photocatalytic process than in the other methods. Using a five-level CCD model, the operational parameters were evaluated and optimized. Based on the ANOVA analysis results, higher determination coefficients (R^2^ = 98.9% and adj-R^2^ = 97.9%) confirmed the accuracy of the obtained model to predict the degradation efficiency of RB5 in sonophotocatalytic process. The main oxidative species in this process were detected to be h ^+^
_VB_ and superoxide radicals. The results revealed that Tb-doped CdSe nanoparticles can be utilized in various experimental cycles with no significant drop in photocatalytic activity.

## Data Availability

The data presented in this study are available on request from the corresponding author.

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
