# Peer review of "Sonochemical Synthesis, Characterization and Optical Properties of Tb-Doped CdSe Nanoparticles: Synergistic Effect between Photocatalysis and Sonocatalysis"

_nanomaterials, 2021, doi:10.3390/nano11020378_

Round 1

Reviewer 1 Report

Abstract

Change “determined” with characterized” and “spices” with “species”

Introduction

Delete “and” before “etc” (and etc [17-20].)

Reformulate the sentence: “The sono and photocatalytic application of pure CdSe nanoparticles has been rarely reported due to the several issues which its narrow band gap is the main.” (Which its narrow band gap is the main ?)

Section 2.2

Change l with Lambda symbol (l=1.5406 AËš)

What is the meaning of “surface state” in the sentence “Using an electron microscope (SEM, S-4200, Hitachi), the surface state and the morphology were observed.”

Section 2.3

Change “was” to “were” in sentence “Next, drops of hydrazine hydrate (N2H4.H2O) was inserted…”

Change ‘the result was’ in the sentence: “The results were a black powder”

Chapter 2.4

Why the drying after catalytic tests was performed at 70C if the initial one was at 50C (“…dried at vacuum for 3 h at 50°C.”). Why not both at 50 or 70?

Results

The authors claims that “Compared to the pure CdSe, the SEM images show larger crystal size of the terbium-doped CdSe nanoparticles.” This do not seems to be in agreement with XRD results. Please clarify this aspect. However, it may be possible that “This proves that the doping of Tb3+ions into the CdSe lattice increases the aggregation of nanoparticles” but this is not related with the crystallization process.

The asymmetric peak in Fig 5c suggests that it may be possible to be composed of (at least) two sub-peaks, as well as the Tb second peak (around 1240 eV). This may imply two chemical states. Please resolve somehow this aspect.

The sentence “Table 4 lists the band gap energy for Tb-doped CdSe and pure” must be reformulated (something like “Table 4 lists the band gap energy of pure and Tb-doped CdSe”)

I do prefer to show in Fig 6 the whole abs. spectrum and the Tauc plot as an inset.

Add a “c” (excite) and an “e” (electrons) in the sentence: “However, the US radiation cannot exite the elctrones”

The decomposition of RB5 without catalyst must be shown (under separate and combined sono- and photo-catalytic tests) in Fig 7a, if possible.

Conclusion

The sense of the first sentence is completely wrong “Through a simple and eco-friendly sonochemical technique, Tb-doped CdSe nanoparticles were synthesized using a new visible-light-responsive photocatalyst” (maybe the authors intended to write “resulting” instead of “using”). Also do not use terms as “new” since the material is not new.

Please reformulate the sentence “ANOVA analysis represented higher determination coefficients”

Correct “oxidative aperies” to “oxidative species”

General comment

The authors are asked to explain if the combination of sono- and photo-catalytic degradation could have a practical application?

Author Response

** Reviewer # 1 

Comments and Suggestions for Authors

Abstract

Change “determined” with characterized” and “spices” with “species”

 Response:

Thank you for the attention. We corrected them.

Introduction

Delete “and” before “etc” (and etc [17-20].)

 Response:

According to the comment, we corrected it.

Reformulate the sentence: “The sono and photocatalytic application of pure CdSe nanoparticles has been rarely reported due to the several issues which its narrow band gap is the main.” (Which its narrow band gap is the main?)

  Response:

According to the comment, the sentence was rewritten as: The sono and photocatalytic application of pure CdSe nanoparticles has been rarely reported due its narrow band gap.

Section 2.2

Change l with Lambda symbol (l=1.5406 AËš)

Response:

Thank you for the comment. It was changed.

What is the meaning of “surface state” in the sentence “Using an electron microscope (SEM, S-4200, Hitachi), the surface state and the morphology were observed.”

 Response:

Based on the comment, we removed “surface state”. It was meaningless here

Section 2.3

Change “was” to “were” in sentence “Next, drops of hydrazine hydrate (N2H4.H2O) was inserted…” Change ‘the result was’ in the sentence: “The results were a black powder”

Response:

Thank you for the comment. It was changed.

Chapter 2.4

Why the drying after catalytic tests was performed at 70C if the initial one was at 50C (“…dried at vacuum for 3 h at 50°C.”). Why not both at 50 or 70?

Response:

Thank you for the comment. We had a typing mistake. For reusability tests, the catalyst was dried at 50°C.

Results

The authors claims that “Compared to the pure CdSe, the SEM images show larger crystal size of the terbium-doped CdSe nanoparticles.” This do not seems to be in agreement with XRD results. Please clarify this aspect. However, it may be possible that “This proves that the doping of Tb3+ions into the CdSe lattice increases the aggregation of nanoparticles” but this is not related with the crystallization process.

Response:

We also agree with the opinion of the honorable referee. We cannot comment about the crystal size of materials because SEM images shows the particles containing several crystal. SEM image indicate the morphology and particle size. We should distinguish between a particle and a crystal. According to the comment, we corrected the sentence as “Compared to the pure CdSe, the SEM images show larger particle size of the terbium-doped CdSe nanoparticles.”  

The asymmetric peak in Fig 5c suggests that it may be possible to be composed of (at least) two sub-peaks, as well as the Tb second peak (around 1240 eV). This may imply two chemical states. Please resolve somehow this aspect.

Response: The sentence modified.

Two intense peaks around 1276.49 and 1243.56 eV are assigned to the Tb 3d5/2 and Tb 3d3/2, respectively according to the references:

  1. A Facile Synthesis and Photoluminescence Properties of SiO2:Tb3+ Spherical Nanoparticles Advances in Nanoparticles, 2017, 6, 11-21.
  2. Growth, structural and thermophysical properties of TbNbO4 crystals, CrystEngCom 20(10)2018. DOI:  10.1039/C7CE02072D.

The sentence “Table 4 lists the band gap energy for Tb-doped CdSe and pure” must be reformulated (something like “Table 4 lists the band gap energy of pure and Tb-doped CdSe”).

Response:

According to the comment the sentence was rewritten as “The band gap energy of pure and Tb-doped CdSe are given in Table 4.”

I do prefer to show in Fig 6 the whole abs. spectrum and the Tauc plot as an inset.

Response: The Absorbance spectra of as-prepared compounds provided in text as Fig.6 and the text refined.

The UV-Vis diffuse reflectance spectrum was used to study the optical absorption property, as shown in Fig. 6. To quantify the shift of the absorption edge, the band-gap energies of Tb-doped CdSe and pure CdSe were calculated by Tauc’s equation.

Add a “c” (excite) and an “e” (electrons) in the sentence: “However, the US radiation cannot exite the elctrones”

Response:

Thank you. We corrected them.

The decomposition of RB5 without catalyst must be shown (under separate and combined sono- and photo-catalytic tests) in Fig 7a, if possible.

Response:

Fortunately, we have the requested experimental results in our archive. Accordingly, the decomposition of RB5 without catalyst results are inserted in Fig. 7.

Conclusion

The sense of the first sentence is completely wrong “Through a simple and eco-friendly sonochemical technique, Tb-doped CdSe nanoparticles were synthesized using a new visible-light-responsive photocatalyst” (maybe the authors intended to write “resulting” instead of “using”).

Response:

Thank you for the comment. We corrected the sentence as ” Through a simple sonochemical technique, Tb-doped CdSe nanoparticles were synthesized and successfully used as a visible-light-responsive sonophotocatalyst.”

Also do not use terms as “new” since the material is not new.

Response:

According to the comment, it was removed.

Please reformulate the sentence “ANOVA analysis represented higher determination coefficients”

Response:

According to the comment, it was rewritten as “Based on the ANOVA analysis results, higher determination coefficients (R2 =98.9% and adj-R2 =97.9%) confirmed the accuracy of obtained model to predict the degradation efficiency of RB5 in sonophotocatalytic process.”

Correct “oxidative aperies” to “oxidative species”

Response:

According to the comment, it was corrected.

 General comment

The authors are asked to explain if the combination of sono- and photo-catalytic degradation could have a practical application?

Response:

Our main aim from this work was the use of CdSe photocatalytic application especially in degradation of the persistent organic pollutants due to the desired properties noted in Introduction section. Then we carried out primary evaluation about its performance in degradation of RB5 as a sample organic pollutant.  However, in the case of the practical application of the sonophotocatalytic processes we believe that the mentioned processes can be applicable even they have the ability to be commercialized in the future. As you know, our prepared catalyst can be activated under 40 W visible light lamp which is very cost effective than common photocatalysts such as TiO2 and ZnO activated by UV illumination. In this regard the solar energy can be exploited. So, if we can design the cost effective ultrasonic reactors, this type of advanced oxidation processes will be applicable. It must be mentioned that other aspects of the photo/sonocatalytic processes such as the mineralization efficiency, cost, sustainability, flexibility, reliability etc. must be evaluated in large scale applications. Undoubtedly, a comprehensive review of a process cannot be done in a thesis or an article. Several founded project are needed and our work the initial steps are to introduce a catalyst for this purpose.     

Reviewer 2 Report

The topic is interesting, the data presented are applicable, and the experiments were properly performed and adequately presented / discussed. This manuscript is suitable for publication in Nanomaterials. However, there are minor corrections needed throughout the manuscript.

Results and discussion. The interpretation of the obtained results/correlations is not entirely performed in relationship with past (similar) studies on the topic. Therefore, the authors are strongly encouraged to introduce some studies related to their work and to correlate the obtained results according to previous studies.

References. There is a tendency of self-citation. This should be reconsidered.

I consider that the article can be accepted for publication only after a minor revision.

Author Response

Comments and Suggestions for Authors

The topic is interesting, the data presented are applicable, and the experiments were properly performed and adequately presented / discussed. This manuscript is suitable for publication in Nanomaterials. However, there are minor corrections needed throughout the manuscript.

Results and discussion. The interpretation of the obtained results/correlations is not entirely performed in relationship with past (similar) studies on the topic. Therefore, the authors are strongly encouraged to introduce some studies related to their work and to correlate the obtained results according to previous studies.

Response:

According to the comment, the photocatalytic/sonocatalytic performance of  prepared Tb-doped CdSe nanoparticles with that of the pristine, doped or some composites of  CdSe in degradation of organic pollutants were given in Table S1.

References. There is a tendency of self-citation. This should be reconsidered.

Response:

Thank you for the comment. We reconsidered the references and remove the self-citations. Only, Refs. 24, 25, 29 and 35 remained from the authors.

I consider that the article can be accepted for publication only after a minor revision.

Reviewer 3 Report

The present work “Sonochemical synthesis, characterization and optical properties of Tb-doped CdSe nanoparticles: Synergistic effect between photocatalysis and sonocatalysis” deals with the sonochemical synthesis of Tb-doped CdSe nanoparticles and the study of their sono-photocatalytic activity for the degradation of Reactive Black 5.

I have serious concerns about this research. In the first part, the synthesis of a novel type of photocatalyst, consisting on Tb-doped CdSe nanoparticles is described. Authors claim that they have developed a simple and eco-friendly synthesis; since they use extremely toxic and carcinogenic and hydrazine, this is totally untrue. Selenium is not near as toxic, but its accumulative effect causes serious health problems.

Anyway, the big problem lays with the experimental design. When you present a novel reagent, you should investigate first the advantages and disadvantages with respect to the previously reported reagents developed for this purpose. This key part is missing in this work. Then, authors move to the study of the degradation of RB5 as a model dye contaminant. The reason why they choose this dye is a mystery to me. Because of the lack of studies about the photocatalytic degradation of this particular dye, a comparison with other catalysts is impossible.

Moreover, it is widely known that the use of ultrasonic baths for sonochemistry can result in reproducibility problems, as the power is not homogeneous in different positions in the bath. When you add photochemistry to the equation, the reproducibility problems are even worse. This had been studied thoroughly by Colmenares and co-workers, but all the Colmenares’s group research is missing in the literature section.

To sum up, due to my concerns about the experimental design and the integrity of the results, I can’t recommend the publication of this work in Nanomaterials.

Author Response

Comments and Suggestions for Authors

The present work “Sonochemical synthesis, characterization and optical properties of Tb-doped CdSe nanoparticles: Synergistic effect between photocatalysis and sonocatalysis” deals with the sonochemical synthesis of Tb-doped CdSe nanoparticles and the study of their sono-photocatalytic activity for the degradation of Reactive Black 5.

I have serious concerns about this research. In the first part, the synthesis of a novel type of photocatalyst, consisting on Tb-doped CdSe nanoparticles is described. Authors claim that they have developed a simple and eco-friendly synthesis; since they use extremely toxic and carcinogenic and hydrazine, this is totally untrue. Selenium is not near as toxic, but its accumulative effect causes serious health problems.

Response:

We removed the eco-friendly from text. Hydrazine in reaction changes to N2 and H2 that is not toxic. Also, just we use 1 mL per each experiment that not affects the atmosphere. Previously many researches done on CdSe compounds as follows:

  1. Heterogeneous sonocatalytic degradation of anazolene sodium by synthesized dysprosium

doped CdSe nanostructures, https://doi.org/10.1016/j.ultsonch.2017.07.021

  1. Kinetic modeling of sonocatalytic performance of Gd-doped CdSe nanoparticles for

degradation of Acid Blue 5. https://doi.org/10.1016/j.ultsonch.2017.04.022

  1. Kinetics and Mechanism of Enhanced Photocatalytic Activity under Visible Light Using

Synthesized Pr x Cd 1−x Se Nanoparticles. dx.doi.org/10.1021/ie402352g | Ind. Eng. Chem. Res.

2013, 52, 13357−13369

Anyway, the big problem lays with the experimental design. When you present a novel reagent, you should investigate first the advantages and disadvantages with respect to the previously reported reagents developed for this purpose. This key part is missing in this work. Then, authors move to the study of the degradation of RB5 as a model dye contaminant. The reason why they choose this dye is a mystery to me. Because of the lack of studies about the photocatalytic degradation of this particular dye, a comparison with other catalysts is impossible.

Response:

The main purpose of our work is that we prepare a novel sono/photocatalyst which exploit the unique properties of the CdSe. This is the general purpose of all published articles about the sono or photocatalyst. Then, the performanve of a catalyst was evaluated in degradation of an organic pollutant such as dyes. There are several published articles examining the prepared catalyst efficiency in degrading Reactive Black 5 as a sample organic pollutant as listed following:

1) Catalysis Today.Volume 209, 15 June 2013, Pages 116-121.

Photocatalytic degradation of Reactive Black 5 with TiO2-coated magnetic nanoparticles.

2) J Water Health (2018) 16 (5): 773–781.

Photocatalytic degradation of reactive black 5 on the surface of tin oxide microrods.

3) Bulletin of Materials Science volume 34, pages551–556(2011).

 Photocatalytic degradation of reactive black-5 dye using TiO2 impregnated ZSM-5.

4) Environ. Sci. Technol. 2007, 41, 16, 5846–5853.

 Mechanism of the Photocatalytic Degradation of C.I. Reactive Black 5 at pH 12.0 Using SrTiO3/CeO2 as the Catalyst.

5)  Egyptian Journal of Chemistry,Volume 63, Issue 4, April 2020, Page 1443-1459.

 Photocatalytic Degradation of Reactive Black 5 Using Photo-Fenton and ZnO Nanoparticles under UV Irradiation.

6) Water Res. 2004 Jun;38(11):2775-81,

The photocatalytic degradation of reactive black 5 using TiO2/UV in an annular photoreactor.

7) Separation Science and Technology Volume 50, 2015 - Issue 9,

.Photocatalytic Degradation of Reactive Black-5 Dye with Novel Graphene-Titanium Nanotube Composite.

8) Journal of Physics: Conf. Series 1221 (2019) 012027,

Reactive Black-5 Photodegradation by TiO2 Thin Films Prepared by Ultrasonic Spray.

9) Nanoscale Research Letters volume 4, Article number: 709 (2009),

Photocatalytic Degradation of Two Commercial Reactive Dyes in Aqueous Phase Using Nanophotocatalysts.

10) Materials Science Forum Vol. 764,

Photocatalytic Degradation of Alizarin Cyanine Green G, Reactive Red 195 and Reactive Black 5 Using UV/TiO2 Process.

11) Water Research,Volume 41, Issue 10, May 2007, Pages 2236-2246,

 Photocatalytic degradation of reactive black 5 in aqueous solutions: Effect of operating conditions and coupling with ultrasound irradiation.

12) Journal of Hazardous Materials, Volume 137, Issue 3, 11 October 2006, Pages 1371-1376, Solar assisted photocatalytic and photochemical degradation of Reactive Black 5

13) J Water Health (2018) 16 (5): 773–781.,

 Photocatalytic degradation of reactive black 5 on the surface of tin oxide microrods.

14) Applied Catalysis B: Environmental, Volume 35, Issue 1, 10 December 2001, Pages L1-L7, Photocatalytic degradation of Reactive Black 5: A comparison between TiO2-Tytanpol A11 and TiO2-Degussa P25 photocatalysts.

15) Materials Science in Semiconductor Processing, Volume 16, Issue 4, August 2013, Pages 1109-1116

Photocatalytic degradation of reactive black 5 azo dye by zinc sulfide quantum dots prepared by a sonochemical method.

16) Journal of Environmental Chemical Engineering, Volume 6, Issue 5, October 2018, Pages 6059-6068

Hierarchical CuO microstructures synthesis for visible light driven photocatalytic degradation of Reactive Black-5 dye.

17) Environmental Science and Pollution Research volume 27, pages17438–17445(2020), Surfactant-assisted synthesis of copper oxide nanorods for the enhanced photocatalytic degradation of Reactive Black 5 dye in wastewater.

18) Materials Chemistry and Physics,Volume 243, 1 March 2020, 122635

Structural properties and photocatalytic degradation efficiency of CuO and erbium doped CuO nanostructures prepared by thermal decomposition of some Cu-salophen type complexes as precursors.

19) Ultrasonics Sonochemistry, Volume 20, Issue 1, January 2013, Pages 386-394

Kinetic investigation on sono-degradation of Reactive Black 5 with core–shell nanocrystal.

20) Ultrasonics Sonochemistry, Volume 34, January 2017, Pages 98-106

Kinetic modeling of sonocatalytic degradation of reactive orange 29 in the presence of lanthanide-doped ZnO nanoparticles.

21) Japanese Journal of Applied Physics, Volume 52, Number 7S

Sonochemical Degradation of Reactive Black 5 with a Composite Catalyst of TiO2/Single-Walled Carbon Nanotubes.

22) Japanese Journal of Applied Physics, Volume 49, Number 7S,

Effects of Salt and pH on Sonophotocatalytic Degradation of Azo Dye Reactive Black 5.

23)  Applied Catalysis B: Environmental, Volume 57, Issue 1, 15 April 2005, Pages 55-62

Photocatalytic degradation of azo-dyes reactive black 5 and reactive yellow 145 in water over a newly deposited titanium dioxide.

Moreover, it is widely known that the use of ultrasonic baths for sonochemistry can result in reproducibility problems, as the power is not homogeneous in different positions in the bath. When you add photochemistry to the equation, the reproducibility problems are even worse. This had been studied thoroughly by Colmenares and co-workers, but all the Colmenares’s group research is missing in the literature section.

Unfortunately, we cannot find the studies of Colmenares and co-workers. However, our working process in the laboratory is that after obtaining the initial result, five samples are synthesized  in the same conditions and the photocatalytic performance of the five prepared samples  in photocatalytic degradation are evaluated for final assurance. Fortunately, these results were available in our archive. Our results indicated that the reproducibility of sono/photocatalytic degradations were acceptable and they can be inserted in the manuscript. Also, to consider the comment, the values of degradation efficiencies were given in following Table as ± confidence limits (at 95% confidence level), according to the following Ref:

Time

First

Second

Third

Fourth

Fifth

Average

0

0

0

0

0

0

0

0

10

21.2

20.83

24.21

25.22

23.01

22.93

22.93±1.75

20

36.45

38.75

39.12

41.21

42.35

39.81

39.81±2.12

30

66.57

65.92

62.93

62.32

61.92

64.01

64.01±1.99

40

70.25

68.25

71.52

75.24

74.23

71.97

71.97±2.64

50

78.21

79.25

80.25

77.52

78.52

78.66

78.66±0.96

60

84.52

83.25

89.65

88.25

89.01

86.94

86.94±2.64

a The results are given as   where  is the mean of n repeats and s is the standard deviation.

It must be mentioned that the results appeared to differ unreasonably from the others (doubtful results) were determined by Q-test [according to following ref.].

 Ref. Statistics and chemometrics for analytical chemistry / James N. Miller and Jane C. Miller. 5th ed., Harlow, England, Pages: 52-54.

To sum up, due to my concerns about the experimental design and the integrity of the results, I can’t recommend the publication of this work in Nanomaterials.

Thanks a lot for the reviewer valuable comments.

Best Regards,

Dr.Younes Hanifehpour

Round 2

Reviewer 1 Report

The authors have made the required corrections. Some typing errors are still present (e.g. "However, the US radiation cannot excite the electrones from conduction band to valance dand of a semiconductor. --> US must be UV, electrones must be electrons, valence dand must be valence band)

Therefore the manuscript must be checked for such mistakes and corrected.

Author Response

Response to reviewer

Comments and Suggestions for Authors

The authors have made the required corrections. Some typing errors are still present (e.g. "However, the US radiation cannot excite the electrones from conduction band to valance dand of a semiconductor. --> US must be UV, electrones must be electrons, valence dand must be valence band)

Therefore the manuscript must be checked for such mistakes and corrected.

Response: Done as suggested. The text also refined again in terms of grammatical errors or typos and mistakes. See the highlighted corrections.
